# Effect of Interbody Implants on the Biomechanical Behavior of Lateral Lumbar Interbody Fusion: A Finite Element Study

**DOI:** 10.3390/jfb14020113

**Published:** 2023-02-17

**Authors:** Hangkai Shen, Jia Zhu, Chenhui Huang, Dingding Xiang, Weiqiang Liu

**Affiliations:** 1China United Engineering Corporation, Hangzhou 310000, China; 2Tsinghua Shenzhen International Graduate School, Tsinghua University, Shenzhen 518055, China; 3School of Mechanical Engineering and Automation, Northeastern University, Shenyang 110057, China; 4State Key Laboratory of Tribology in Advanced Equipment, Tsinghua University, Beijing 100084, China

**Keywords:** finite element, porous scaffold, lumbar interbody fusion, osteoporosis, endplate stress

## Abstract

Porous titanium interbody scaffolds are growing in popularity due to their appealing advantages for bone ingrowth. This study aimed to investigate the biomechanical effects of scaffold materials in both normal and osteoporotic lumbar spines using a finite element (FE) model. Four scaffold materials were compared: Ti6Al4V (Ti), PEEK, porous titanium of 65% porosity (P65), and porous titanium of 80% porosity (P80). In addition, the range of motion (ROM), endplate stress, scaffold stress, and pedicle screw stress were calculated and compared. The results showed that the ROM decreased by more than 96% after surgery, and the solid Ti scaffold provided the lowest ROM (1.2–3.4% of the intact case) at the surgical segment among all models. Compared to solid Ti, PEEK decreased the scaffold stress by 53–66 and the endplate stress by 0–33%, while porous Ti decreased the scaffold stress by 20–32% and the endplate stress by 0–32%. Further, compared with P65, P80 slightly increased the ROM (<0.03°) and pedicle screw stress (<4%) and decreased the endplate stress by 0–13% and scaffold stress by approximately 18%. Moreover, the osteoporotic lumbar spine provided higher ROMs, endplate stresses, scaffold stresses, and pedicle screw stresses in all motion modes. The porous Ti scaffolds may offer an alternative for lateral lumbar interbody fusion.

## 1. Introduction

Lumbar interbody fusion has been widely used in the treatment of lumbar degenerative diseases [1]. Due to the large footprint area, lateral lumbar interbody fusion had the advantage of significantly reducing the range of motion (ROM) of the index segment compared to other surgical techniques [2]. Traditional titanium scaffolds could reduce the ROM effectively and achieve a satisfactory fusion rate [1,3]. However, previous studies reported that titanium scaffolds may result in complications such as stress shielding, scaffold migration, and scaffold subsidence because of the high stiffness of the solid titanium [4,5]. Polyetheretherketone (PEEK) scaffolds have shown suitable elastic modulus and excellent clinical performance [6]. However, PEEK scaffolds showed a higher risk of instability (up to 32.8%) because of their poor cell adhesion effect [7].

Recently, porous titanium scaffolds manufactured by additive manufacturing (AM) have also been used in lumbar interbody fusion. By adjusting pore topology and porosity, the porous AM scaffold could achieve optimal mechanical properties [8,9]. In addition, the porous scaffold facilitated bone ingrowth, which was beneficial for long-term stability [10]. Moreover, the porous structure was similar to that of the cancellous bone, which could promote nutrient transportation [11]. The interlock between the porous structure and bony elements could also reduce the risk of scaffold migration [12]. Some studies have investigated the biomechanical performance of porous scaffolds using finite element (FE) and experimental methods. For example, Meena et al. [13] reported that porous scaffolds with pore sizes ranging from 400–600 μm could significantly decrease the stress shielding. Similarly, Tsuruga et al. [14] also reported that pore sizes of 300–400 μm were optimum for bone ingrowth. In the numerical and experimental study of Tsai et al. [15], they reported that a porosity of 69–80% achieved satisfactory biomechanical performances. Wang et al. [9] also optimized the design of porous scaffolds using methods of topology optimization.

In addition, the prevalence of osteoporosis has increased dramatically among the elderly in recent years, and an increasing number of patients requiring lumbar interbody fusion suffer from osteoporosis [16]. Previous studies reported that osteoporosis significantly altered the normal biomechanics of the lumbar spine [17] and showed a higher risk of fracture, internal fixation failure, and scaffold subsidence [18,19]. However, according to the existing literature, the biomechanics of lateral scaffolds with various materials were not fully understood. Further, few studies have investigated the impact of scaffold materials on the biomechanical performance of the osteoporotic lumbar spine. It was necessary to study the postoperative biomechanics of the osteoporotic lumbar spine.

Consequently, the aim of this study was to evaluate the biomechanical effects of scaffold materials in both the normal and osteoporotic lumbar spines using the finite element (FE) method. The main novelty of the present study is the application of porous scaffolds in both the normal and osteoporotic lumbar spines. Four scaffold materials were compared: PEEK, Ti6Al4V, 65% porous Ti, and 80% porous Ti. The material properties of porous Ti were measured by the compression test. A total of 10 FE models were developed and simulated. The results of ROM, endplate stress, scaffold stress, and pedicle screw stress were calculated and simulated.

## 2. Materials and Methods

### 2.1. Finite Element Model of the Lumbar Spine

A previously developed and validated FE model of the intact L1-5 human lumbar spine was used in this study (Figure 1) [20]. The geometry of the current model was obtained from 0.7-mm-thick computed tomography (CT) scans of a healthy volunteer (female, age 37 years, height 158 cm, weight 52 kg). Then a total of 492 CT images were transformed into a 3D geometric lumbar model, which was meshed in Hypermesh (Version 14.0, Altair Technologies, Inc., Fremont, CA, USA). The current model is composed of cortical bone (1 mm thick) [21], cancellous bone, posterior bone, intervertebral disc, and all seven kinds of ligaments: anterior longitudinal ligament, posterior longitudinal ligament, ligament flava, interspinous ligament, supraspinal ligament, intertransverse ligament, and capsular ligament. The intervertebral disc included a cartilaginous endplate (0.5 mm thick), an annulus fibrosus, and a nucleus pulposus [21]. The ligaments were simulated by tension-only truss elements. The frictionless surface-to-surface contact was applied between the facet joints [22]. Finally, the intact L1-5 FE model included 120,978 nodes and 555,063 elements, which could effectively ensure the accuracy of the calculation. Abaqus (Version 14.0, Simulia, Inc., Providence, RI, USA) was used to perform the FE simulation.

### 2.2. Finite Element Models of L3-4 Lateral Interbody Fusion Construct

In order to simulate the lateral interbody fusion (LIF) procedure, the total nucleus pulposus and partial annulus were removed at the L3-4 segment [23]. The scaffold was inserted at the L3-4 disc space with bilateral pedicle screw fixation (Figure 2). PEEK, titanium alloy (Ti6Al4V, Ti), and two titanium scaffolds with varying porosities (65% and 80%, P65, and P80) were simulated as lateral scaffold materials. [23,24]. All the scaffolds were filled with graft bone to promote bone ingrowth. The porous scaffolds were produced through 3D printing technology, with an average pore size of 350–400 μm [23,24]. All the scaffolds were 28-mm long, 9-mm wide, and 7-mm high. The pedicle screws (45 mm in length and 6 mm in diameter) were interconnected by longitudinal rods (6 mm in diameter). The bone-screw and bone-scaffold interfaces were simulated as tie constraints to represent the long-term effects after instrumentation [25]. The material of the pedicle screws was Ti6Al4V. The porous scaffolds and pedicle screws used in this study were created using Solidworks 2014 software (Dassault Systemes SolidWorks Corp., Waltham, MA, USA).

Figure 3a displays the mechanical test for the porous materials. A compression test was performed based on the ISO13314:2011 standard [23,24] by using a universal material testing system (Instron 8874, Instron Corporation, Canto, MA, USA). The compression test was carried out according to the displacement-controlled protocol [26]. The loading rate during the test was set at 1 mm/min. The load-displacement curves were recorded, and the Young’s modulus of the porous titanium materials was obtained. The average Young’s modulus was 8800 MPa for the P65 samples and 5000 MPa for the P80 samples.

In order to further investigate the impact of the scaffold materials on the osteoporotic lumbar spine, the osteoporotic lumbar FE model was also developed. According to the previous studies [27,28], the Young’s modulus of all bony structures was reduced, by 66% for the cancellous bone, and by 33% for the cortical bone endplates, and posterior elements. The thickness of the cortical bone was not significantly changed [29]. Table 1 summarizes the material properties used in the FE models.

Additionally, to validate the intact lumbar model, we followed the same protocol used in the previous study [26]. The inferior surface of L5 was constrained in all directions, and then four moments (8 Nm flexion, 6 Nm extension, 6 Nm lateral bending, and 4 Nm axial rotation) were applied to the L1 segment, respectively. The simulated results of the FE model were compared with the experimental results of Renner et al. [26].

Further, we used six human cadaver lumbar spines (L1-5) to further validate the FE model (Figure 3b). The lumbar spines were obtained from the Research Institute of Tsinghua University in Shenzhen, and the experiment followed an institutional medical ethics procedure. The L5 segment of the lumbar spine specimens were fixed by Wood’s alloy. A compressive follower load of 280 N, which represented partial body weight, was applied along the curvature of the lumbar spine, and then four moments (8 Nm flexion, 6 Nm extension, 6 Nm lateral bending, and 4 Nm axial rotation) were applied at the L1 segment through the MTS Bionix Spine Simulation Machine (MTS system, Inc., Minneapolis, MN, USA), respectively. The experiment was performed at a room temperature of 23 °C and a humidity of 35–80%. In addition, 0.9% saline was sprayed on the specimen periodically to keep it moist. Each segment of the lumbar spine was implanted with an NDI marker (NDI, Inc., Waterloo, ON, Canada), which consisted of four targets. The location of each segment was represented by the coordinates of the NDI markers. The rate at the moment was 0.1°/s. Each motion mode (flexion, extension, lateral bending, and axial rotation) was repeated three times, and the results of the ROM were measured and averaged for analysis. Finally, the results of the ROM of each segment were compared with the results of the intact FE model in order to validate the FE model.

### 2.3. Boundary and Loading Conditions

In this study, apart from the intact model, four lateral scaffold materials were simulated: PEEK, Ti6Al4V (Ti), 65% porous Ti (P65), and 80% porous Ti (P80). The inferior surface of the L5 vertebral body was fixed in all directions, and a follower load of 280 N was applied along the contour of the lumbar spine to represent the body weight [21]. Then, a moment of 7.5 Nm was applied at the L1 vertebral body to simulate flexion, extension, lateral bending, and axial rotation [21,24]. Both the normal and osteoporotic spine conditions were perform. In total, 40 simulation calculations for 10 models (five normal models and five osteoporotic models) and four motion modes were performed. The following results were simulated and measured: ROM at the surgical segment, endplate stress, scaffold stress, and pedicle screw stress.

## 3. Results

### 3.1. Model Validation

Under 280 N follower load and 7.5 Nm moment, the simulated results of L1-5 ROM were compared with the experimental results of Renner et al. [26]. These results were also compared with the results of our cadaveric specimen test, as shown in Figure 4. The total ROMs of the intact L1-5 lumbar spine were 34.4° in flexion-extension, 13.4° in axial rotation, and 30.8° in left-right lateral bending. The predicted ROM of each segment was within one standard deviation of the results derived from previous studies [26] and our cadaveric measurement. Therefore, the current FE model was validated.

### 3.2. ROM of the Surgical Segment

Under 280 N follower load and 7.5 Nm moment, the ROM of the surgical segment was displayed in Figure 5. The ROMs of the surgical models were normalized to the intact model [23]. For the normal spine condition (Figure 5a), the predicted ROMs for all surgical models decreased by more than 96% after the lateral scaffold was inserted. ROMs of the solid Ti scaffold were the lowest compared among all the scaffolds (1.2–3.4% of the intact model). ROMs of the P65 scaffold were slightly lower than those of the P80 scaffold (deviation within 0.03°). The PEEK scaffold and the P80 scaffold showed similar ROMs in all motion modes.

For the osteoporotic spine condition, the ROMs of the surgical models decreased by more than 94% compared with the intact model (Figure 5b). Compared with the normal spine, the ROMs were higher in the osteoporotic spine, and the trend of the ROMs was similar.

### 3.3. Endplate Stress

Figure 6 displays the endplate stress of the L3 inferior endplate. For the normal spine condition, the Ti scaffold provided the highest endplate stress among all surgical models (Figure 6a). The PEEK and P80 scaffolds appeared to provide similar endplate stresses in all motion modes. Compared with P80, the endplate stress of P65 was slightly higher in extension, lateral bending, and axial rotation. For the osteoporotic spine condition, the endplate stress increased by 3.4–15.9% compared with the normal spine condition (Figure 6b). The Ti scaffold provided the highest endplate stress in extension and axial rotation. However, in flexion and lateral bending, PEEK and P80 showed higher endplate stresses.

Figure 7 displays the endplate stress of the L4 superior endplate. For both normal and osteoporotic spine conditions, the Ti scaffold provided the highest endplate stress in all motion modes. Compared with P80, P65 provided higher endplate stresses. Compared with the normal spine condition, the endplate stress increased by 3.0–25% for osteoporotic spines.

### 3.4. Scaffold Stress

Figure 8 showed the maximum scaffold stresses of the surgical models. For both normal and osteoporotic spines, the Ti scaffold provided the highest scaffold stress in all motion modes, followed by P65 and P80, while PEEK had the lowest scaffold stress (Figure 8a). Compared to solid Ti, PEEK decreased the scaffold stress by 53–66%, and porous Ti decreased the scaffold stress by 20–32%. In addition, P80 decreased the scaffold stress by approximately 18% compared with P65.

Compared with normal spines, osteoporotic spines increased the maximum scaffold stress by 13–30% in flexion, 13–22% in extension, 12–29% in lateral bending, and 20–28% in axial rotation. In addition, as shown in Figure 8b, the maximum scaffold stress always appeared at the edges of the scaffold.

### 3.5. Pedicle Screw Stress

Figure 9 showed the maximum pedicle screw stresses of the surgical models. For the Ti scaffold, the pedicle screw stress was slightly lower than that of PEEK, P65, and P80, but the difference was not statistically significant. Compared with normal spines, osteoporotic spines increased the maximum pedicle screw stress by 6–13% in flexion, 6–7% in extension, 28–31% in lateral bending, and 14–17% in axial rotation. In addition, as shown in Figure 9b, the maximum pedicle screw stress appeared at the joint between the rod and screw.

## 4. Discussion

The current study investigated the biomechanical effect of lateral scaffolds made of various materials in both normal and osteoporotic lumbar spines using the FE method. Four scaffold materials were evaluated: solid Ti, PEEK, P65, and P80, of which the material properties of the porous Ti were measured via mechanical testing. Such finite element research was not feasible in a human cadaveric experiment because of the difficulty in measuring soft tissue stresses. The predicted ROM of each segment was comparable to a previous study [26] and our cadaveric measurement. Additionally, the effects of the scaffold materials on endplate stress, scaffold stress, and pedicle screw stress were also investigated. We believe this study may be helpful in understanding the role of porous Ti scaffolds in lateral lumbar interbody fusion.

As displayed in Figure 5, the ROM of the surgical segment decreased by more than 90% in all motion modes compared with the intact model, and the Ti scaffold offered the least ROM. In addition, the ROM of P80 was slightly higher compared with P65. These findings were consistent with the FE research of Zhang et al. [23], who reported that the ROM increased as the porosity increased. Compared with the normal spine, ROMs were higher in the osteoporotic spine, but the differences were not significant. This finding indicated that osteoporosis did not significantly alter the ROM of the lumbar spine, similar to the findings of previous studies [29]. This was due to the fact that the intervertebral disc of the osteoporotic spine did not degenerate significantly, and its mobility was not markedly changed [31].

Scaffold subsidence was associated with the stresses at the bone-scaffold interface [24]. As shown in Figure 6 and Figure 7, the porous Ti scaffolds showed advantages in lowering the endplate stresses. This was because the stress shielding was inversely proportional to the stiffness of the scaffold [4], and the porous Ti provided a Young’s modulus more comparable to bone elements. Therefore, the porous scaffold may decrease the risk of scaffold subsidence. Due to its lower mechanical strength, the osteoporotic spine had a higher risk of scaffold subsidence [32]. The current study found that the endplate stresses increased significantly in osteoporotic spines, consistent with the findings in the research of Zhang et al. [33]. Furthermore, compared to solid Ti, PEEK, P65, and P80, they decreased the endplate stresses of the osteoporotic spine in extension, lateral bending, and axial rotation, indicating a lower risk of subsidence. In addition, PEEK and P80 showed similar endplate stresses this may be because the Young’s moduli of the two materials were similar.

A previous study has reported that the increased scaffold stress might result in subsidence and scaffold failure [32]. Porous Ti had advantages in reducing cage stress compared to solid Ti, and P65 had a greater cage stress than P80. These findings were consistent with the biomechanical research of Zhang et al. [23]. In addition, compared with the normal spine, the scaffold stress increased by 12–30% in osteoporotic spines. Similarly, Liu et al. [29] also reported that the stresses on the scaffolds of patients with osteoporosis were slightly higher than those of a normal spine. These findings indicated that the osteoporotic bone had a higher risk of scaffold failure, and the PEEK and porous Ti scaffolds might decrease the risk of scaffold failure, especially in osteoporotic spines.

As shown in Figure 9, the solid Ti scaffold slightly decreased the pedicle screw stress compared with PEEK, P65, and P80. This was because the solid Ti scaffold provided the highest stiffness in flexion, extension, lateral bending, and axial rotation. In addition, we found that the maximum pedicle screw stress appeared at the joint between the rod and screw. This finding agreed with the results of Fan et al. [34], who reported that the stress was concentrated in the neck regions of the pedicle screws and the middle regions of the rods. In addition, we also found that the osteoporotic bone had a higher risk of pedicle screw breakage.

In summary, the porous Ti scaffolds showed many advantages over the solid Ti scaffolds and may be an alternative to PEEK scaffolds in lateral lumbar interbody fusion. In addition to its favorable biomechanical performances, porous Ti also showed advantages in facilitating bone ingrowth, promoting nutrient transportation, and reducing the risk of scaffold migration because of its micro-pore structure. A recent prospective study (40 patients with 53 segments) reported that the bone fusion rates for porous Ti scaffolds ranged from 86.7–94.3% [35]. Additionally, a recent study compared the clinical outcomes of porous Ti versus PEEK scaffolds for stand-alone lateral lumbar interbody fusion [36], and the results showed that patients in the porous Ti group had a significantly lower overall subsidence rate (20% vs. 58.8%, P = 0.004) and revision rate. Moreover, patients in the porous Ti group showed significantly greater pain reduction. In the biocompatibility evaluation (porcine model) of Tsai et al. [37], they found high porosities, especially 60 and 80%, facilitated more bone formation inside the implant, and bone formation was higher in the porous Ti group than in the PEEK group. This may be explained by the poor cell adhesion effect of the PEEK scaffolds [12]. Similarly, McGilvray et al. [38] also found that porous Ti significantly increased bone ingrowth over PEEK in their ovine lumbar fusion model. Recently, a number of porous Ti cages have been commercially available, but there have been limited long-term clinical trials supporting their efficacy in promoting early osseointegration and fusion [39]. Although increasing the porosity may be more beneficial to bone growth, the mechanical strength of the porous Ti should be taken into account [24]. Therefore, it required careful consideration to further increase the porosity.

Although the lateral scaffolds showed favorable clinical outcomes and advantages in stabilizing the surgical segment, the high incidence of adjacent segment degeneration (ASD) should be taken into consideration. According to a previous review of research, the incidence of radiographic ASD ranged from 5.2% to 100% after lumbar interbody fusion [40]. This may be because the stress and motion of adjacent segments have increased extremely after lumbar fusion [23,24]. In our future study, we will focus on the biomechanical performances of the adjacent segments, and optimize the structure of the lateral scaffolds to mitigate ASD through preserving spinal movement and improving load transmission in the lumbar spine.

In addition, the current research performed the FE simulation in static loading conditions (280 N follower load + four pure moments). However, patients may be inevitably exposed to dynamic loads in daily life, such as the whole-body vibration caused by vehicles [20]. Guo et al. [41] investigated the stress responses of the lumbar spine under dynamic loads and reported that, compared with static loads, dynamic loads significantly increased the intradiscal pressure by 242.4%. This finding indicated that dynamic loads may cause more damage to the lumbar spine compared to static loads. Therefore, the next task for our team is to investigate the dynamic biomechanical performances of porous Ti scaffolds.

The limitations in the current FE study should be pointed out. Firstly, because only one unique FE model was employed, the simulated results might not be representative of an average person. Secondly, the porous scaffolds were simplified as “solid-mimic” scaffolds rather than porous structures, and the mechanical properties were estimated from the mechanical test. However, many previous studies have adopted this simplification to improve computing efficiency [23,24]. In addition, the same follower load was applied for the lumbar spine, although there were some changes in muscle activities after lateral lumbar interbody fusion. However, the tendency of the predicted results would not be significantly changed by these simplifications.

## 5. Conclusions

The present research attempted to investigate the biomechanical effects of scaffold materials in both the normal and osteoporotic lumbar spines. The solid Ti scaffold showed advantages in stabilizing the surgical segment and decreasing the risk of pedicle screw breakage. However, solid Ti showed higher risks of scaffold breakage, scaffold subsidence, and stress shielding. PEEK and porous Ti showed significant advantages over solid Ti in decreasing endplate stresses and scaffold stresses, especially in osteoporotic spines. Compared with the normal spine, the osteoporotic lumbar spine provided higher ROMs, endplate stresses, scaffold stresses, and pedicle screw stresses in all motion modes. The porous Ti scaffolds may offer an alternative for lateral lumbar interbody fusion. Further clinical studies are necessary to validate the findings of this study.

## Figures and Tables

**Figure 1 jfb-14-00113-f001:**
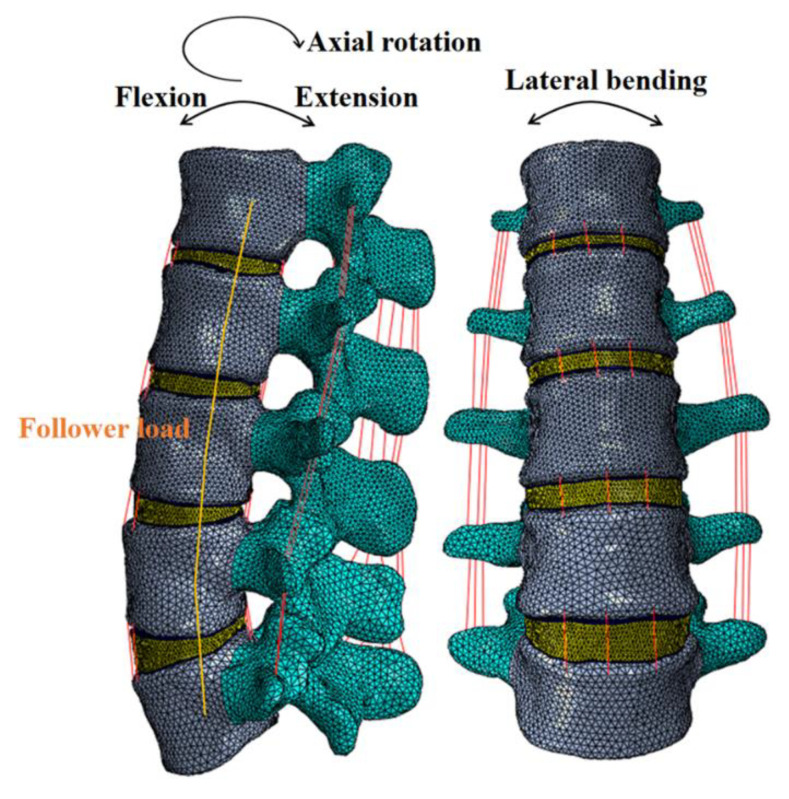
Finite element model of the L1-5 lumbar spine.

**Figure 2 jfb-14-00113-f002:**
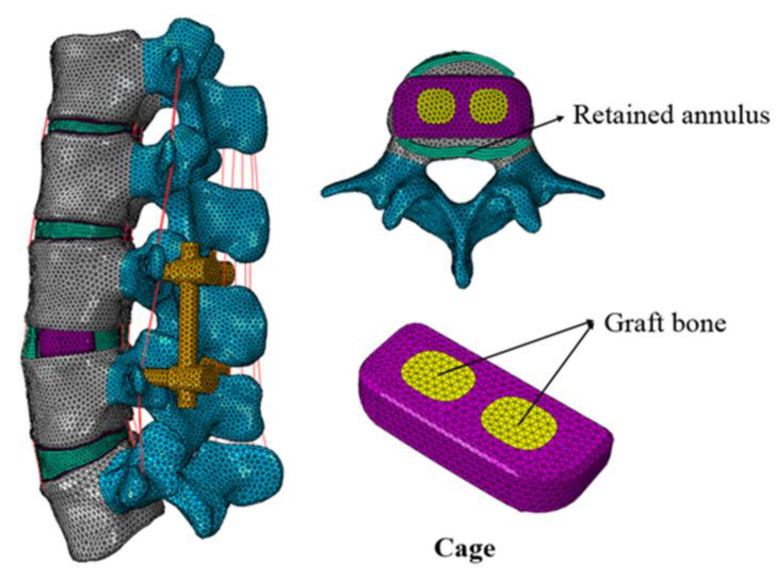
Finite element model of L3-4 lateral interbody fusion.

**Figure 3 jfb-14-00113-f003:**
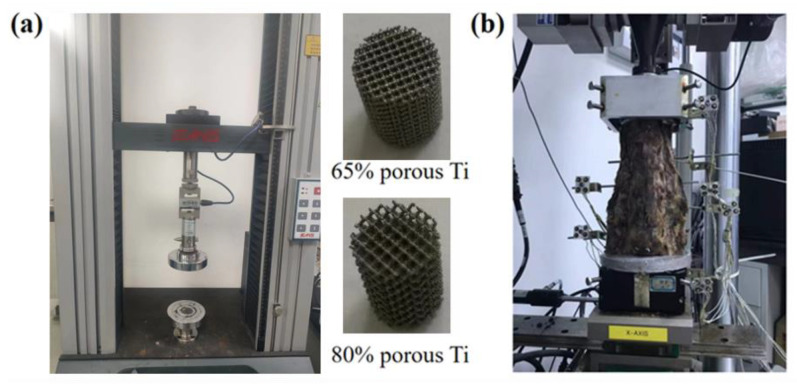
(**a**) Mechanical tests for the porous materials; (**b**) In vitro test with 6 human cadaver lumbar spines.

**Figure 4 jfb-14-00113-f004:**
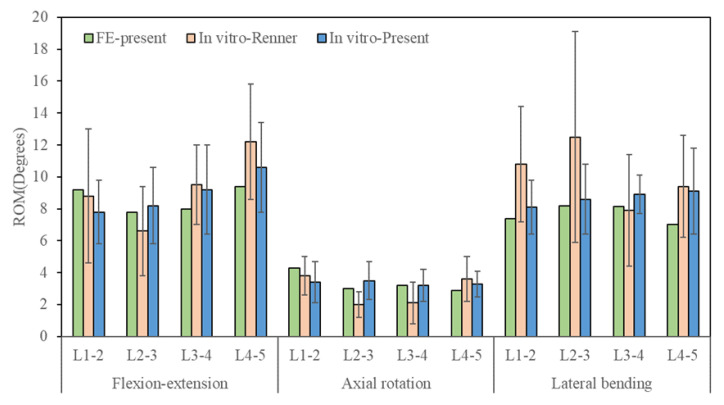
Validation of the finite element model.

**Figure 5 jfb-14-00113-f005:**
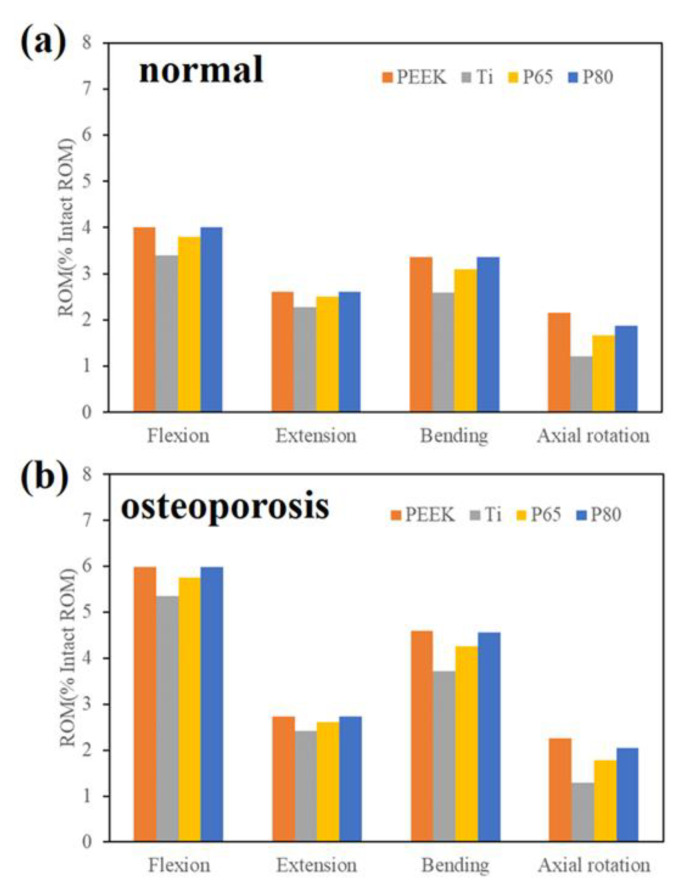
ROM of the surgical segment. (**a**) Normal spine condition; (**b**) Osteoporotic spine condition.

**Figure 6 jfb-14-00113-f006:**
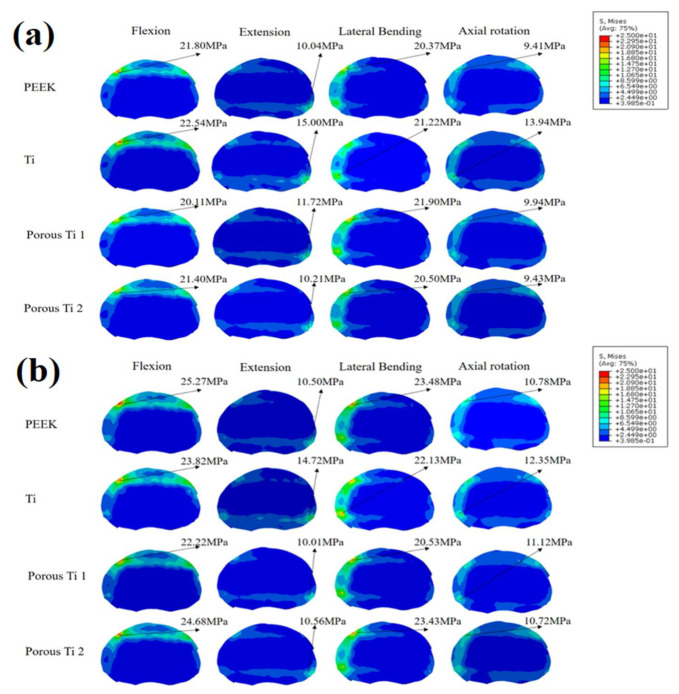
L3 inferior endplate stress. (**a**) Normal spine condition; (**b**) Osteoporotic spine condition.

**Figure 7 jfb-14-00113-f007:**
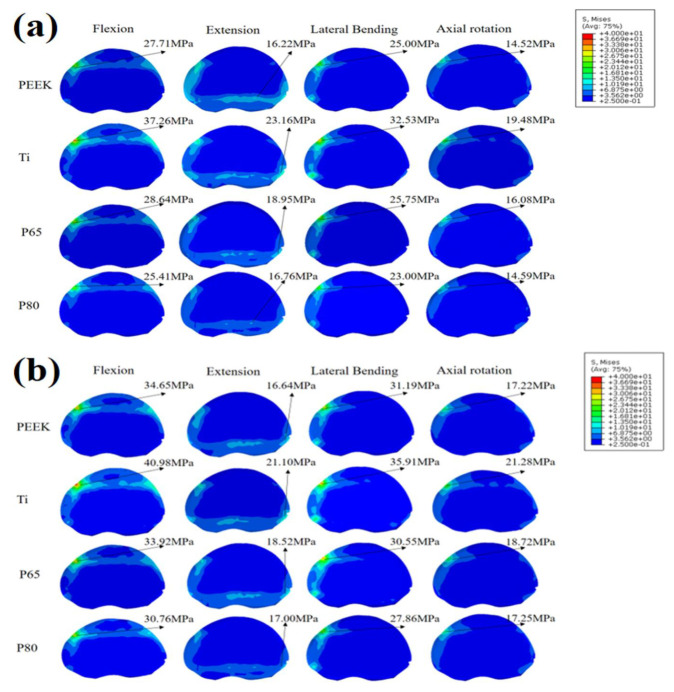
L4 superior endplate stress. (**a**) Normal spine condition; (**b**) Osteoporotic spine condition.

**Figure 8 jfb-14-00113-f008:**
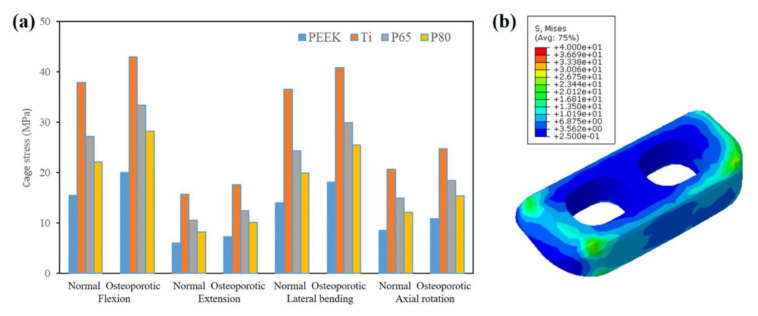
(**a**) The maximum scaffold stress of the surgical models; (**b**) The maximum scaffold stress appeared at the edges of the scaffold.

**Figure 9 jfb-14-00113-f009:**
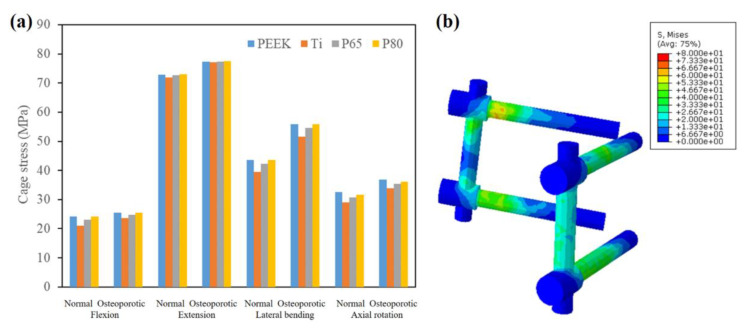
(**a**) The maximum pedicle screw stress of the surgical models; (**b**) The maximum pedicle screw stress appeared at the joint between rod and screw.

**Table 1 jfb-14-00113-t001:** Material properties used in the FE models.

Component	Young’s Modulus (MPa)	Poisson Ratio	Cross-Sectional Area (mm^2^)	References
Cortical bone	12,000 (osteoporosis:8040)	0.3	-	[23]
Cancellous bone	100 (osteoporosis:34)	0.2	-	[23]
Posterior bone	3500 (osteoporosis:2345)	0.25	-	[21]
Endplate	24 (osteoporosis:16.1)	0.25	-	[30]
Nucleus pulposus	1	0.49	-	[23]
Annulus fibrosus	4.2	0.45	-	[23]
Anterior longitudinal	20	0.3	63.7	[21]
Posterior longitudinal	20	0.3	20	
Ligament flava	19.5	0.3	40	
Interspinal	11.6	0.3	40	
Supraspinal	15	0.3	30	
Intertransverse	58.7	0.3	3.6	
Capsular	32.9	0.3	60	
PEEK	3500	0.3	-	[23]
Ti6Al4V	110,000	0.3	-	[23]
65% porous Ti	8800	0.05	-	Experiment
80% porous Ti	5000	0.05	-	Experiment
Graft Bone	3500	0.25	-	[23]

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
