# Peer review of "Effect of Interbody Implants on the Biomechanical Behavior of Lateral Lumbar Interbody Fusion: A Finite Element Study"

_jfb, 2023, doi:10.3390/jfb14020113_

Round 1

Reviewer 1 Report

Ref. No.: jfb-2115329

Subject: Decision on Manuscript: Effect of interbody implants on the biomechanical behavior of 2 lateral lumbar interbody fusion: a finite element study

Journal: Journal of Functional Biomaterials

Dear Editor,

I would like to thank you for the invitation to collaborate to review process of article “Effect of interbody implants on the biomechanical behavior of 2 lateral lumbar interbody fusion: a finite element study”. I recommend that is necessary a major revision of manuscript. Some comments are described below:

English should be improved in all manuscript.

Abstract: Changing “effects of cage materials in both normal a….” to “effects of scaffolds in both normal a....”, similar to other mentions to word cage. In addition, more quantitative information should be added to the abstract.

Introduction: Similar to abstract, all term cages should be replaced to scaffolds. The use of the term scaffold is more appropriate to biomaterials.

In addition, the novelty of this research is not so clear in the introduction. The differences of this study with others in the literature should be emphasized.

Page 3 Line 106: “To validate the intact lumbar model, we followed the same protocol used in the previous studies [26].” Was more than one study, because the reference is only one?

Figure 5, Figure 8 and 9 should be included error bar.

Moreover, statistical analysis beyond deviation should be performed.

Figures 6 and 7 should be improved the resolution.

Figure 8b the legend of colors should be included. Similar to Figure 9b.

Discussion should be in-depth.

Conclusion should be re-written.

Author Response

Reviewer #1:I would like to thank you for the invitation to collaborate to review process of article “Effect of interbody implants on the biomechanical behavior of 2 lateral lumbar interbody fusion: a finite element study”. I recommend that is necessary a major revision of manuscript. Some comments are described below:

Comment 1: English should be improved in all manuscript.

Response 1: Thank you for your careful review and valuable suggestions. The English of this manuscript has been polished and highlighted.

Comment 2: Abstract: Changing “effects of cage materials in both normal a….” to “effects of scaffolds in both normal a....”, similar to other mentions to word cage. In addition, more quantitative information should be added to the abstract.

Response 2: Thank you for your careful review and valuable suggestions. The word “cage” has been changed to “scaffolds”. In addition, we have revised the Abstract and added more quantitative information as:              

“The results showed that the ROM decreased by more than 96% after surgery, and the solid Ti scaffold provided the lowest ROM (1.2-3.4% of the intact case) at the surgical segment among all models. Compared to solid Ti, PEEK decreased the scaffold stress by 53-66%, and the endplate stress by 0-33%, while porous Ti decreased the scaffold stress by 20-32%, and the endplate stress by 0-32%. Compared with P65, P80 slightly increased the ROM (<0.03°) and pedicle screw stress (<4%), and decreased the endplate stress by 0-13% and scaffold stress by approximately 18%. Moreover, the osteoporotic lumbar spine provided higher ROMs, endplate stresses, scaffold stresses and pedicle screw stresses in all motion modes.”

 Comment 3: Introduction: Similar to abstract, all term cages should be replaced to scaffolds. The use of the term scaffold is more appropriate to biomaterials.

Response 3: Thank you for your careful review and valuable suggestions. As the reviewer recommended, all term cages have been replaced to scaffolds.

 Comment 4: In addition, the novelty of this research is not so clear in the introduction. The differences of this study with others in the literature should be emphasized.

Response 4: Thank you for your careful review and valuable suggestions. We have revised the novelty of this research as:“However, according to the existing literature, the biomechanics of lateral scaffolds with various materials were not fully understood. In addition, few studies have investigated the impact of scaffold materials on biomechanical performance of the osteoporotic lumbar spine. It was necessary to study the postoperative biomechanics of the osteoporotic lumbar spine. Therefore, the aim of this study was to evaluate the biomechanical effects of scaffold materials in both the normal and osteoporotic lumbar spines by finite element (FE) method.” 

Comment 5: Page 3 Line 106: “To validate the intact lumbar model, we followed the same protocol used in the previous studies [26].” Was more than one study, because the reference is only one?

Response 5: Thank you for your careful review. There was only one reference, and this sentence has been revised as “To validate the intact lumbar model, we followed the same protocol used in the previous study”. 

Comment 6: Figure 5, Figure 8 and 9 should be included error bar. Moreover, statistical analysis beyond deviation should be performed.

Response 6: Thank you for your careful review. Figures 5, 8, and 9 were derived from the results of finite element analysis, the results represented the exact values rather than the average values. Therefore, we followed the same method of the previous studies [1-5], and the error bar was not included in these figures.        In addition, the predicted ROM of each segment (FE result) was within one standard deviation of the results derived from previous experimental studies, so statistical analysis beyond deviation was not performed.  

Comment 7: Figures 6 and 7 should be improved the resolution.

Response 7: Thank you for your valuable advice, the resolution of Figures 6 and 7 has been improved. 

Comment 8: Figure 8b the legend of colors should be included. Similar to Figure 9b.

Response: Thank you for your valuable advice. Figures 8b and 9b have been improved and the legend of colors has been be included. 

Comment 9: Discussion should be in-depth.

Response: Thank you for your valuable advice. We have revised the discussion and these parts were highlighted.

 Comment 10: Conclusion should be re-written.

Response: Thank you for your valuable advice. The conclusion has been re-written as:“The present research attempted to investigate the biomechanical effects of scaffold materials in both the normal and osteoporotic lumbar spines. The solid Ti scaffold showed advantages in stabilizing the surgical segment and decreasing the risk of pedicle screw breakage. However, solid Ti showed higher risks of scaffold breakage, scaffold subsidence and stress shielding. PEEK and porous Ti showed significant advantages over solid Ti in decreasing endplate stresses and scaffold stresses, especially in osteoporotic spines. Compared with the normal spine, the osteoporotic lumbar spine provided higher ROMs, endplate stresses, scaffold stresses and pedicle screw stresses in all motion modes. The porous Ti scaffolds may offer an alternative for lateral lumbar interbody fusion. Further clinical studies are necessary to validate the findings of this study.”

 Thanks again for your careful review and kind suggestion!

References: [1] Zhang Z, Li H, Fogel GR, Xiang D, Liao Z, Liu W: Finite element model predicts the biomechanical performance of trans-foraminal lumbar interbody fusion with various porous additive manufactured scaffolds. Computers in Biology and Medicine 2018, 95:167-174.

[2] Fan W, Guo L-X, Zhang M: Biomechanical analysis of lumbar interbody fusion supplemented with various posterior stabi-lization systems. European Spine Journal 2021, 30(8):2342-2350.

[3] Renner SM, Natarajan RN, Patwardhan AG, Havey RM, Voronov LI, Guo BY, Andersson GBJ, An HS: Novel model to analyze the effect of a large compressive follower pre-load on range of motions in a lumbar spine. Journal of Biomechanics 2007, 40(6):1326-1332.

[4] Liu Z-X, Gao Z-W, Chen C, Liu Z-Y, Cai X-Y, Ren Y-N, Sun X, Ma X-L, Du C-F, Yang Q: Effects of osteoporosis on the bio-mechanics of various supplemental fixations co-applied with oblique lumbar interbody fusion (OLIF): a finite element analysis. Bmc Musculoskeletal Disorders 2022, 23(1).

[5] Rastegar S, Arnoux P-J, Wang X, Aubin C-E: Biomechanical analysis of segmental lumbar lordosis and risk of scaffold sub-sidence with different scaffold heights and alternative placements in transforaminal lumbar interbody fusion. Computer Methods in Biomechanics and Biomedical Engineering 2020, 23(9):456-466.

Reviewer 2 Report

Very interesting paper, would only recommend that the authors ,can dwell more on the outcome of different type of cage material used and backed it up more by citing clinical reference if found, because this is a Finite element study.

The main question the authors are checking in this paper is the difference between the different materials used for lumbar surgery. There are no clear data about the real difference between Titanium, peek, or porous cages in terms of fusion postop, rate of subsidence, and so on, hence, leaving the surgeon to rely on personal experience and opinion.  The subject is very interesting in helping us spine surgeons understand the real difference and taking the topic out of the marketing aspect.  My only comment is to add more citations and references regarding real clinical data in terms of the 4 different types of cages used.

Author Response

Reviewer #2:

Comment: Very interesting paper, would only recommend that the authors, can dwell more on the outcome of different type of cage material used and backed it up more by citing clinical reference if found, because this is a Finite element study.

The main question the authors are checking in this paper is the difference between the different materials used for lumbar surgery. There are no clear data about the real difference between Titanium, peek, or porous cages in terms of fusion postop, rate of subsidence, and so on, hence, leaving the surgeon to rely on personal experience and opinion. The subject is very interesting in helping us spine surgeons understand the real difference and taking the topic out of the marketing aspect.  My only comment is to add more citations and references regarding real clinical data in terms of the 4 different types of cages used.

Response:Thank you very much for your careful review and kind suggestion!Although a number of porous Ti cages have been commercially available, there were limited long-term clinical trials supporting their efficacy in promoting early osseointegration and fusion. Therefore, we believe this finite element study was helpful for understanding the biomechanics of lateral fusion cages with various materials.

As the reviewer suggested, we have added the following part:

“In addition to its favorable biomechanical performances, porous Ti also showed advantages in facilitating bone ingrowth, promoting nutrient transportation, and reducing the risk of scaffold migration because of its micro-pore structure. A recent prospective study (40 patients with 53 segments) reported that the bone fusion rates for porous Ti scaffolds ranged from 86.7-94.3% [1]. Additionally, a recent study compared the clinical outcomes of porous Ti versus PEEK scaffolds for stand-alone lateral lumbar interbody fusion [2], and the results showed that patients in the porous Ti group had a significantly lower overall subsidence rate (20% vs. 58.8%, P = 0.004) and revision rate. Moreover, patients in the porous Ti group showed significantly greater pain reduction. In the biocompatibility evaluation (porcine model) of Tsai et al. [3], they found high porosities, especially 60 and 80%, facilitated more bone formation inside the implant, and bone formation was higher in the porous Ti group than in the PEEK group. This may be explained by the poor cell adhesion effect of the PEEK scaffolds [4]. Similarly, McGilvray et al. [5] also found that porous Ti significantly increased bone ingrowth than PEEK in their ovine lumbar fusion model. Recently, a number of porous Ti cages have been commercially available, but there were limited long-term clinical trials supporting their efficacy in promoting early osseointegration and fusion [6]. Although increasing the porosity may be more beneficial to bone growth, the mechanical strength of the porous Ti should be taken into account [7]. Therefore, it required careful consideration to further increase the porosity.” 

Thanks again for your kind suggestion! 

References:

[1] Chung SS, Lee KJ, Kwon YB, Kang KC: Characteristics and Efficacy of a New 3-Dimensional Printed Mesh Structure Titanium Alloy Spacer for Posterior Lumbar Interbody Fusion. ORTHOPEDICS 2017, 40(5): 880-885.

[2] Amini DA, Moser M, Oezel L, Zhu J, Okano I, Shue J, Sama AA, Cammisa FP, Girardi FP, Hughes AP: Early Outcomes of Three-Dimensional–Printed Porous Titanium versus Polyetheretherketone Cage Implantation for Stand-Alone Lateral Lumbar Interbody Fusion in the Treatment of Symptomatic Adjacent Segment Degeneration. World neurosurgery 2022, 162: 14-20.

[3] Tsai PI, Wu MH, Li YY, Lin TH, Tsai JSC, Huang HI, Lai HJ, Lee MH, Chen CY: Additive-manufactured Ti-6Al-4V/Polyetheretherketone composite porous cage for Interbody fusion: bone growth and biocompatibility evaluation in a porcine model. BMC MUSCULOSKELETAL DISORDERS, 22(1).

[4] Makino T, Takaneka S, Sakai Y, Yoshikawa H, Kaito T: Impact of mechanical stability on the progress of bone ongrowth on the frame surfaces of a titanium-coated PEEK scaffold and a 3D porous titanium alloy scaffold: in vivo analysis using CT color mapping. European Spine Journal 2021, 30(5):1303-1313.

[5] McGilvray KC, Easley J, Seim HB, Regan D, Berven SH, Hsu WK, Mroz TE, Puttlitz CM: Bony ingrowth potential of 3D-printed porous titanium alloy: a direct comparison of interbody cage materials in an in vivo ovine lumbar fusion model. SPINE JOURNAL 2018, 18(7): 1250-1260.

[6] Toop N, Gifford C, Motiei-Langroudi R, Farzadi A, Boulter D, Forghani R, Farhadi HF: Can activated titanium interbody cages accelerate or enhance spinal fusion? a review of the literature and a design for clinical trials. JOURNAL OF MATERIALS SCIENCE-MATERIALS IN MEDICINE 2021, 33(1).

[7] Zhang Z, Li H, Fogel GR, Xiang D, Liao Z, Liu W: Finite element model predicts the biomechanical performance of trans-foraminal lumbar interbody fusion with various porous additive manufactured scaffolds. Computers in Biology and Medicine 2018, 95:167-174.

Reviewer 3 Report

This is an interesting, well written study on the effect o different cage materials on lumbar interbody fusion, using a finite element analysis.

Author Response

thank you very much for your careful review and kind suggestion!

Round 2

Reviewer 1 Report

Ref. No.: jfb-2115329-v2

Subject: Decision on Manuscript: Effect of interbody implants on the biomechanical behavior of 2 lateral lumbar interbody fusion: a finite element study

Journal: Journal of Functional Biomaterials

Dear Editor,

I would like to thank you for the invitation to collaborate to review process of article “Effect of interbody implants on the biomechanical behavior of 2 lateral lumbar interbody fusion: a finite element study”. My recommendation is described below:

The authors did the required all corrections and the manuscript is publishable in current version.
